# Tuberculosis-Associated Hemophagocytic Lymphohistiocytosis: A Review of Current Literature

**DOI:** 10.3390/jcm12165366

**Published:** 2023-08-18

**Authors:** Trym Fauchald, Bjørn Blomberg, Håkon Reikvam

**Affiliations:** 1Faculty of Medicine, University of Bergen, 5007 Bergen, Norway; trym.fauchald@student.uib.no; 2Department of Clinical Science, University of Bergen, 5007 Bergen, Norway; bjorn.blomberg@uib.no; 3Department of Medicine, Haukeland University Hospital, 5021 Bergen, Norway

**Keywords:** tuberculosis, hemophagocytic lymphohistiocytosis, hemophagocytic syndrome, cytokines, ferritin, tuberculostatic

## Abstract

Hemophagocytic lymphohistiocytosis (HLH) is a condition of immune dysregulation and hyperinflammation, leading to organ failure and death. Malignancy, autoimmune conditions, and infections, including Mycobacterium tuberculosis (TB), are all considered triggers of HLH. The aim of this study was to review all reported cases of TB-associated HLH in English literature, and to summarize the epidemiology, diagnostics, treatment, and mortality in patients with concomitant HLH and TB. A systematic review of described cases with TB-associated HLH, via a structured literature search in the medical database PubMed, is presented. Additional articles were included through cross-referencing with existing review articles. Articles were reviewed based on a predetermined set of criteria. A total of 116 patients with TB-associated HLH were identified with a male:female ratio of about 3:2. The age at presentation ranged from 12 days to 83 years. Malignancy, autoimmunity, and renal failure were the most common comorbid conditions. Most patients received both tuberculostatic and specific immunomodulating treatment, which was associated with a 66% (48/73) survival rate compared to 56% (15/27) in those receiving only tuberculostatic treatment, and 0% (0/13) in those receiving only immunomodulating treatment. The survival rate was 55% overall. The overlapping presentation between disseminated TB and HLH poses challenging diagnostics and may delay diagnosis and treatment, leading to increased mortality. TB should be considered as a potential trigger of HLH; clinicians’ knowledge and awareness of this may result in the appropriate investigations needed to ensure diagnosis and proper treatment.

## 1. Introduction

Hemophagocytic lymphohistiocytosis (HLH) is a disorder of dysfunctional immune regulation characterized by an excessive activation of macrophages and lymphocytes, resulting in systemic inflammation and tissue destruction. It is characterized clinically by fever, organomegaly, cytopenias, hyperferritinemia, hypertriglyceridemia, hypofibrinogenemia, and hemophagocytosis in the reticuloendothelial system. HLH could be classified as either familial, where most cases are seen in infants and small children, or sporadic, which can occur at all ages. Infection is a common trigger in both familial and sporadic cases; other triggers for sporadic cases include malignancy and autoimmune disorders [1,2]. While the pathophysiological immune response is regarded as being largely similar in both familial and sporadic HLH, it is important to identify potential triggers of HLH as treatment of the underlying conditions can be sufficient for full recovery [3,4,5,6,7].

Tuberculosis (TB) was the 13th leading cause of death worldwide in 2019, and was only surpassed by COVID-19 in number of deaths by a single infectious cause in 2020 [8]. The disease is caused by Mycobacterium tuberculosis and has innumerable clinical presentations. Morbidity and mortality rates worldwide are still high, especially in low- and middle-income countries [9]. HLH is a rare, although potentially fatal complication of TB, with reported survival rates of 40–60% [6,10]. Triggering infections associated with HLH are acknowledged in 25% of cases [11]. TB is implicated in 9–25% of HLH related to underlying infection [11,12,13]. Timely diagnosis and treatment are paramount for patient survival; due to the numerous, and often non-specific presentations, of both TB and HLH, this is often difficult. Awareness of HLH among treating health professionals in relation to diseases such as TB is important to be able to diagnose it in a timely manner.

For the past two decades, the diagnosis of HLH has been based on the enrolment criteria for the HLH-04 study [14], which subsequently have become the de facto diagnostic criteria for the condition (Appendix A). This set of criteria includes, among other analyses, the level of soluble interleukin 2-receptor (sIL-2r) and natural killer (NK)-cell activity, the analyses of which are often not readily available. While other criteria have been suggested, including HScore (Appendix A), a validated scoring system with more differentially weighted criteria [15,16], most official guidelines still primarily refer to the criteria of the HLH-04 trial.

In this study, a systematic search on PubMed with a review of all cases of HLH secondary to TB available in English, published between January 1980 and June 2021, was performed. While other reviews have been published, the last review in a major international journal was published in 2006 with 36 reported cases of TB-associated HLH [10]. The aim of this study is to review all reported cases of TB-associated HLH in the English literature, including the previously reviewed cases, and to summarize the epidemiology, diagnostics, treatment, and mortality in patients with concomitant HLH and TB. Given high mortality rates and lack of clear consensus regarding treatment of the coexistence of these two conditions, an updated review with more and recently described cases may be of great value.

## 2. Materials and Methods

Due to the rarity of TB complicated by HLH, there is a lack of clinical consensus on how this clinical entity is best handled. By compiling available articles describing cases of TB-associated HLH, this review aims to give an overview of relevant clinical presentations, diagnostics, and treatments.

A structured literature search was performed in the online database PubMed including articles published up until 1 June 2021. The search combined the following terms: “Tuberculosis” (Mesh) or tuberculosis or synonyms for tuberculosis (Table 1), and “Lymphohistiocytosis, Hemophagocytic” (Mesh) or hemophagocytic lymphohistiocytosis or synonyms for this (Table 1).

This search yielded 208 articles. The process of exclusion and inclusion is summarized in Figure 1: Articles were first excluded based on one or more of the following: the article was not in English, it did not regard specific patient cases, and the patient(s) described did not have suspected TB and HLH concurrently. After this, a total of 91 articles remained. Cross-referencing review articles found in this search yielded eight additional articles.

The remaining articles were evaluated regarding the following:Symptoms and findings supportive of a diagnosis of HLH in accordance with established diagnostic criteria (Appendix A).Diagnosis of MTB based on positive culture, finding of acid-fast bacilli on microscopy, or positive polymerase chain reaction (PCR).

As nine of the articles mentioned more than one patient case, the total number of cases reviewed was 116.

## 3. Results

### 3.1. Epidemiology

The patient group in this study consisted of 116 individual patient cases, of which there were 69 males (60%) and 45 females (39%). In two cases, gender was not specified. The age ranged from 14 days to 83 years, with a median of 40 years (Table 2). In 61 of the cases (52%), no comorbidities were mentioned (Table 3). A total of 74 (64%) patients were given both tuberculostatic treatment and immune-targeted treatment, 24 (21%) patients received tuberculostatic treatment and no immune-targeted treatment, 3 (3%) patients received immune-targeted treatment and no tuberculostatic treatment, and 11 (9%) patients received neither (Table 4). Supportive treatment was described in a total of 63 cases (54%) (Table 4). The outcome for the patients was survival in 63 cases, and death in 52 cases, resulting in an overall survival rate of 54% (Appendix A). The survival rate was higher among patients <30 years of age, ranging from 67% to 100%, compared to older patients (Figure 2).

### 3.2. Clinical Presentation and Findings

The main clinical and biochemical findings are summarized in Table 2. Fever was the most frequently described symptom, presenting in 114 (98%) of cases, followed by hemophagocytosis and bi-/pancytopenia, presenting in 106 (91%) and 103 (89%) cases, respectively. The temperature measured was specified in 60 of the patients, ranging from 37.3 °C to 40.0 °C, with a median of 39.0 °C.

### 3.3. Comorbidity

The number of instances of different comorbidities described in the study population is summarized in Table 3. In 61 (53%) of the cases, no comorbidities were mentioned, and 18 (16%) patients had more than one comorbidity. There were 21 instances of lifestyle-associated conditions, 11 (9%) patients with renal failure, 10 (9%) patients with malignancy—8 with hematologic malignancy and 2 with other malignancies. Ten (9%) patients had some kind of autoimmune disease while eight (7%) patients had a concomitant infection—HIV/AIDS in two of them. Two (2%) female patients were pregnant. There were 15 instances of other comorbidities.

### 3.4. Diagnostic Clinical Criteria

The most frequently used clinical criteria for diagnosing HLH are the eight criteria from the HLH-04 study [14], where having five or more fulfilled criteria out of the eight is considered diagnostic.

In this study, fever was the most frequently fulfilled criterion, followed by the presence of hemophagocytosis, and bi- or pancytopenia (Table 2). Hyperferritinemia >500 µg/L and high levels of soluble IL-2 receptor were described in a large proportion of the patients in which these parameters were mentioned (Table 2). Splenomegaly was described in 79/96 (82%) patients. A total of 66 out of the 74 patients (89%) in which triglycerides and/or fibrinogen were specified fulfilled the criteria for either hypertriglyceridemia, hypofibrinogenemia, or both. NK-cell activity was described as low or absent in 13/16 (81%) patients.

In total, 78 patients fulfilled at least five criteria (67%), including 2 patients who fulfilled all eight criteria. A total of 38 patients fulfilled <5 criteria, and 99 patients fulfilled at least 4 criteria (85%). The number of criteria fulfilled by patients ranged from two to eight, with five being the most common. Appendix A provides an overview of each of the criteria for every patient in this study [3,4,5,6,7,10,13,17,18,19,20,21,22,23,24,25,26,27,28,29,30,31,32,33,34,35,36,37,38,39,40,41,42,43,44,45,46,47,48,49,50,51,52,53,54,55,56,57,58,59,60,61,62,63,64,65,66,67,68,69,70,71,72,73,74,75,76,77,78,79,80,81,82,83,84,85,86,87,88,89,90,91,92,93,94,95,96,97,98,99,100,101,102,103,104,105,106,107,108].

### 3.5. Diagnosis of TB

In all cases included in this review, a diagnosis of TB was made by the respective authors of the articles reviewed. The means of diagnosing TB was either by culture, acid-fast staining, PCR, interferon-gamma release assay, or in some cases made by radiological and/or histopathological evidence and the startling effect of anti-tuberculous therapy (ATT).

Tuberculosis was mainly diagnosed based on microbiological and/or histopathological evidence; the number of times different organ systems were implicated in TB disease is summarized in Figure 3. The main organ system affected was the respiratory tract, as evidenced by microbiological confirmation of TB in samples from sputum, bronchoalveolar lavage (BAL), lung biopsy, pleural effusion, and pleura biopsy in a total of 66 (57%) patients. Bone marrow was the second most common organ involved, from which TB was isolated in a total of 49 (42%) patients. Apart from this, the most common sites of TB were the liver, spleen, and lymph nodes with 16 (14%), 12 (10%), and 23 (20%) cases, respectively. Evidence of TB involvement in ≥2 organ systems was demonstrated in 55 (47%) cases. In 18 (16%) cases, signs of TB were found only in the pulmonary system, and in 10 (9%) cases, TB was found only in the bone marrow. In 73 (63%) patients, TB was considered disseminated, either by the respective case report authors’ descriptions, or by the nature that TB was considered present in multiple organs. In four cases, no specific organ location was described [95,103,106].

A microbiological diagnosis by either culture, demonstration of acid-fast bacilli, or PCR was achieved in 90 cases (78%). Histopathological findings, without microbiological evidence, were used for diagnosis in 15 cases (13%). In 10 patients (9%), the method of diagnosis of TB was not disclosed.

One patient, a two-year-old female, had no reported microbiological or histopathological evidence for TB [76]. She responded well to ATT and had a positive ELISA IgM with negative IgG and IgA, leading to the diagnosis of acute TB infection.

### 3.6. Treatment and Outcome

ATT was specified in 113/116 patients (Table 4 and Appendix A). A total of 100/113 (89%) patients received ATT, and 13/113 (12%) patients did not. In all cases where ATT was not given, the patients succumbed to their illness; in two of these cases, immune-directed therapy was given [37,64]. Immunomodulating therapy was given to 77/115 (67%) patients (Table 4 and Appendix A), 38/115 (33%) did not receive any immunomodulating therapy, and in 1/116 (1%) patients, there was no information regarding treatment [106]. The relative frequency of different medications used as ATT, and medications used as part of immunomodulating therapy, are visualized in Figure 4.

Overall, 64/116 (55%) patients survived (Appendix A). Among the cases where ATT was given, but not immune-directed therapy, 15/27 (56%) patients survived, whereas in the cases where both ATT and immune-directed therapy were given, 48/73 (66%) survived (OR 1.53, CI 0.61, 3.61, *p* = 0.4).

Supportive treatment, including admission to an intensive care unit (ICU), was mentioned in 64 (55%) patient cases. In this group, 37/64 (58%) patients survived. The most common methods of supportive treatment were mechanical/invasive ventilation, and blood transfusion, each with 28 instances. A further summary of supportive treatment modalities is described in Table 4.

Of the 41 patients who received both ATT, immune-guided therapy, and any supportive treatment, 29 (71%) survived, while among the patients who received ATT and supportive treatment, without immune-guided therapy, 7/15 (47%) survived (OR 2.70, CI 0.78, 9.61, *p* = 0.1).

## 4. Discussion

HLH triggered by TB is a rare, although serious and feared condition, associated with a high degree of morbidity and mortality. We have carried out a comprehensive and systematic compilation of the up-to-date and available literature that has been published on this topic.

The survival rate in this study population was 55%, similar to earlier reviews on TB-HLH [5,6,10]. Compared to some reviews on special patient populations, this differs, though these reviews entail only a few patients, with reviews on neonates and infants with TB-HLH, where 6/9 (67%) patients survived [39], and in patients under 20 years of age, where 9/11 (82%) patients survived [22].

Of the 116 cases reviewed, 69 (59%) were male, 45 (39%) were female, and 2 (2%) were of unspecified gender (Table 2). Similar ratios of males to females have been reported in earlier reviews of TB-associated HLH [6,10]. The higher proportion of males in this study population (Figure 2) may be due to a higher burden of TB disease in adult males compared to females, and that gaps in detection and reporting are higher among males [9]. The sex ratio of our study population differs from other reviews, for instance, on CMV and HLH [109], HLH, and biological disease-modifying anti-rheumatic drugs [85]. A study of the epidemiology of HLH in China found that lymphoma-associated HLH was more prevalent among males, whereas HLH due to rheumatic and immune disorders was more prevalent in females [110]. Additionally, this study found no statistically significant difference between male and female prevalence of infection-associated HLH.

In the reviews conducted by Brastianos et al. [10] and Padhi et al. [6], comorbidities were known in 20/36 (55%) patients and 41/63 (65%) patients, respectively. Whereas in the review by Shea et al. there were comorbid conditions in 42% of cases [5]. In this study, comorbid conditions were mentioned in 55/116 (47%) of cases. Even though there is some variation in the rate of comorbid conditions overall, the pattern of the most common conditions remains similar among this and earlier reviews: renal failure/end-stage renal disease being the most common (*n* = 11), followed by malignancy (*n* = 10) and autoimmune conditions (*n* = 10).

The diagnosis of HLH is often difficult, due to unspecific early findings. Fever and malaise are common early symptoms of the syndrome. The initial presentation is often interpreted as an infection or sepsis. Many patients are started on broad-spectrum antibiotics quite early after admission [32,47]. One common feature of patients with HLH is a progressive worsening despite treatment. Characteristic findings in HLH include cytopenias, organomegaly, hypertriglyceridemia, hypofibrinogenemia, and striking hyperferritinemia, as well as low NK-cell activity and high levels of soluble IL2-r. The specific criteria for diagnosing HLH are mentioned above (Appendix A). In the patient population in this study, five (*n* = 36) fulfilled criteria was the most common, followed by six (*n* = 28) and four (*n* = 21). A total of 79 of the patients described fulfilled five or more criteria. The cut off for diagnosis of HLH is five criteria, when using HLH-04 criteria. This has been demonstrated to equate to a low sensitivity and high specificity for the diagnosis [16].

In addition to the eight HLH-04 criteria used for diagnosing HLH, serum levels of C-X-C motif chemokine ligand (CXCL9) have been introduced as an added criterion [1,111]. CXCL9 is an interferon (IFN)-γ-induced chemokine, the levels of which correlate with the IFN-γ activity in HLH [112], and it has been shown to aid differentiation between HLH and sepsis in pediatric patients [113]. It is, however, also implicated in severe TB infections [114]. CXCL9 was mentioned in only one case reviewed in this study [21]. This case displayed a serum level of CXCL9 more than 17 times higher than the upper normal limit. In this case, the patient would have fulfilled 4/9 criteria if CXCL9 was included.

Both ferritin and fibrinogen are acute-phase reactants, and while serum ferritin levels are often strikingly high in patients with HLH, fibrinogen levels are often low [115]. The ferritin levels in HLH are often >1000 µg/L and can reach >10,000 µg/L [85,116]; cases reviewed in this study also exhibit this finding (Table 2). Increased levels of ferritin are one of the more specific findings, with a reported sensitivity of 90% and specificity of 96% at levels above 10,000 µg/L in children [117]. In adults, however, a high ferritin level (>10,000 μg/L) has been shown to be non-specific for HLH [116], still, such a high level of ferritin should raise suspicion of a severe inflammatory or hematological condition [116].

Even though cytopenias often were present at admission, several cases presented with normal blood counts, and developed cytopenias during disease. This course is exemplified in the case presented by Trovik et al. [17], in which the patient had normal blood counts at admission, which developed into pancytopenia; other cases also exhibited this, developing either bicytopenia or pancytopenia [34,37,47,63,86]. The HLH-04 thresholds for anemia and thrombocytopenia were more commonly achieved in this patient population than neutropenia. Leukopenia (WBC < 4.1 × 10^9^/L) was quantitatively described in 68 of the patients reviewed. Despite HLH being associated with cytopenias, leukocytosis (>9.8 × 10^9^/L) was seen at some point in 14 patients. The presence of leukocytosis should therefore not skew clinicians away from a diagnosis of infection-associated HLH.

Hemophagocytosis is most often described in bone marrow aspirates and biopsies, even though hemophagocytosis can be found throughout the reticuloendothelial system [14,118]. In the cases studied in this review, most instances of hemophagocytosis were found in bone marrow. Additionally, hemophagocytosis was described in the liver [97], spleen [31,97], and peripheral blood [49,73].

In the development of the HScore, the significance of LDH was investigated. It was found to not be independently associated with a positive diagnosis of HLH (*p* = 0.72 for levels between 500 and 2000 IU/L and *p* = 0.19 for levels >2000 IU/L, as compared to the reference <500 IU/L) [15]. LDH is, hence, not included in the HScore, despite many clinicians regarding this as an important analysis in the evaluation of HLH [15,119]. This is not to say that an elevated LDH may not ever be a sign of HLH, although, in and of itself, this does not differentiate between HLH and non-HLH in individuals in whom a diagnosis of HLH is considered. In this review, the median LDH in the 53 patients where this was reported was 1144 IU/L (range 247–10 646 IU/L) with the number of patients having an LDH reported >500 IU/L being 46/53 (87%).

Even though it is common for authors to describe lymphadenopathy as being associated with HLH, it is not considered part of the diagnostic tools of HLH-04 criteria or the HScore. As part of the development of HScore, Hejblum et al. [119] performed a web-based Delphi study to determine helpful criteria in diagnosing HLH in adults. As a part of this work, available publications on HLH in adults were reviewed regarding identifying criteria that were being used to diagnose HLH. Lymphadenopathy was not included in this list of 26 criteria that were assessed [119]. Lymphadenopathy should be considered a sign of other underlying conditions in addition to possible HLH in patients where HLH is suspected based on clinical and biochemical findings [1].

Debaugnies et al. performed a study to evaluate the performance of the HScore, in adult and in pediatric patients, as compared to using the HLH-04 criteria [16]. In their study, they found that a lower cut-off value for the diagnosis of HLH could be used to achieve the same specificity as Fardet et al. To achieve a more comparable sensitivity to the one Fardet et al. achieved, Debaugnies et al. had to lower the cut-off value for the HScore [15,16]. Debaugnies et al. concluded in their study that HScore performs better than HLH-04 at the initial presentation, and that when patients deteriorated, the difference in performance diminished when comparing the two [16]. Despite studies showing the superior performance of HScore when compared to HLH-04 [15,16], there were only a few cases in this study where HScore was utilized [13,96].

Naha et al. [78] pointed out in their case report that the clinical and biochemical findings in their patient could have been ascribed readily to TB alone. Their patient fulfilled six HLH-04 criteria and had an HScore of 258 (>98% likelihood of HLH). Thus, there is reason to entertain the possibility of the significant underreporting of TB-associate HLH, especially in resource-constrained regions [78].

Diagnosing tuberculosis can also present difficulties due to the many ways in which it can present, and the non-specific findings that usually appear early in the course of the disease. Though mycobacterial culture remains the gold standard for diagnosing TB, and is needed for comprehensive characterization of drug resistance, the WHO recommends the use of nucleic acid amplification tests (NAATs)/polymerase chain reaction (PCR), for the rapid diagnosis of TB and rifampicin-resistance [120]. NAAT is promoted over acid-fast staining due to NAATs’ higher sensitivity and specificity, and the ability to rapidly identify rifampicin-resistance [120]. National [121] and international guidelines [122] reflect this, recommending always acquiring samples for culture, as well as NAAT, with the goal of early diagnosis and to determine resistance. In addition, guidelines suggest empirical treatment of TB when clinicoradiological suspicion is high, despite negative microbiological tests. As shown in this summary, a microbiological diagnosis was secured in nearly 90% of the patient cases, reflecting that WHO guidelines are followed more often than not in cases of TB-HLH.

Earlier analyses have shown a correlation between higher age and higher mortality rates in patients with HLH [12,85,123,124,125]. Other factors shown to correlate with a worse prognosis in adult patients with sporadic HLH are underlying lymphoma [123,124,125,126,127], lower platelet count [12,123,124,125,126,128], elevated aminotransferase, and elevated lactate dehydrogenase [124].

In one study, treatment with etoposide was associated with a better prognosis [124]. Furthermore, studies allude to early diagnosis and timely treatment being correlated with improved survival, although statistically significant associations are not always shown [126,129]. One study of HLH patients in an ICU setting found all of the interventions mechanical ventilation, vasopressor, and renal replacement therapy to be independently correlated with higher mortality in univariate analysis [126]. The same study found a correlation between underlying B-cell lymphoma and lower mortality, which the authors suggest is possibly related to the use of cancer chemotherapy in these patients, which may be considered treatment both for the precipitating factor and for HLH [126]. Other studies have also found a correlation between lymphoma-associated HLH and a better prognosis when compared to non-malignancy-associated HLH [130].

Several cases displayed a course where either ATT alone or immune-directed therapy alone was started, with little or no tangible improvement in the patients’ conditions. Cases where ATT was initially given, with insufficient response, include patients with autoimmune conditions [55], patients with TB found in bone marrow [61,66], and patients with malignant conditions [83]. Treatments that were added in these patients include IVIG [55,61,66,86], TPE [55,69,83], splenectomy [66], and Janus-kinase inhibitors [20,86]. Two patients were initially treated with ATT, steroid pulse therapy, and hemodialysis and did not improve with this regime [69,83], but started improving after starting TPE in both cases and continuous hemofiltration in one of the cases [69]. Several patients treated initially according to either the HLH-94 or the HLH-04 protocol did not display significant improvement until proper ATT was given as well [70,71,74,92]. In some of these patients, TB was isolated from reticuloendothelial organs [70,71,92].

In one patient treated with both ATT and immune-targeted therapy, full recovery was not achieved until after a splenectomy was performed, which showed both necrotizing granulomas and hemophagocytosis [31].

## 5. Conclusions

Through the methods employed in this study, we have described the patients with concomitant TB and HLH and summarized the common findings and clinical management. The clinical findings commonly associated with HLH were recurring in this population, although leukopenia was less common than in patients with HLH in general. Most patients were treated with both tuberculostatic and chemoimmunotherapy in some capacity, with a large variety of medications being used. The overall survival rate was slightly above 50%, with timely diagnosis and treatment being seen as crucial factors for positive outcomes. Due to the limitations of this study design, causative relationships cannot be established. Both the severity of this entity, and its elusiveness regarding diagnosis may contribute to a high mortality rate, as well as leading to possible underdiagnosis. Patients who did not receive ATT died in all reviewed cases, and the survival rate was numerically higher in patients receiving both ATT and chemoimmunotherapy than those only receiving ATT, although, the difference did not reach statistical significance. Further research aimed toward a better understanding of how these diseases interact with each other, in addition to targeted research regarding optimal treatment regimens, may further improve the prognosis of these patients in the future. Prompt treatment following standard protocols for tuberculostatic treatment and immunomodulation in these patients should be considered in order to improve outcomes in these patients.

## Figures and Tables

**Figure 1 jcm-12-05366-f001:**
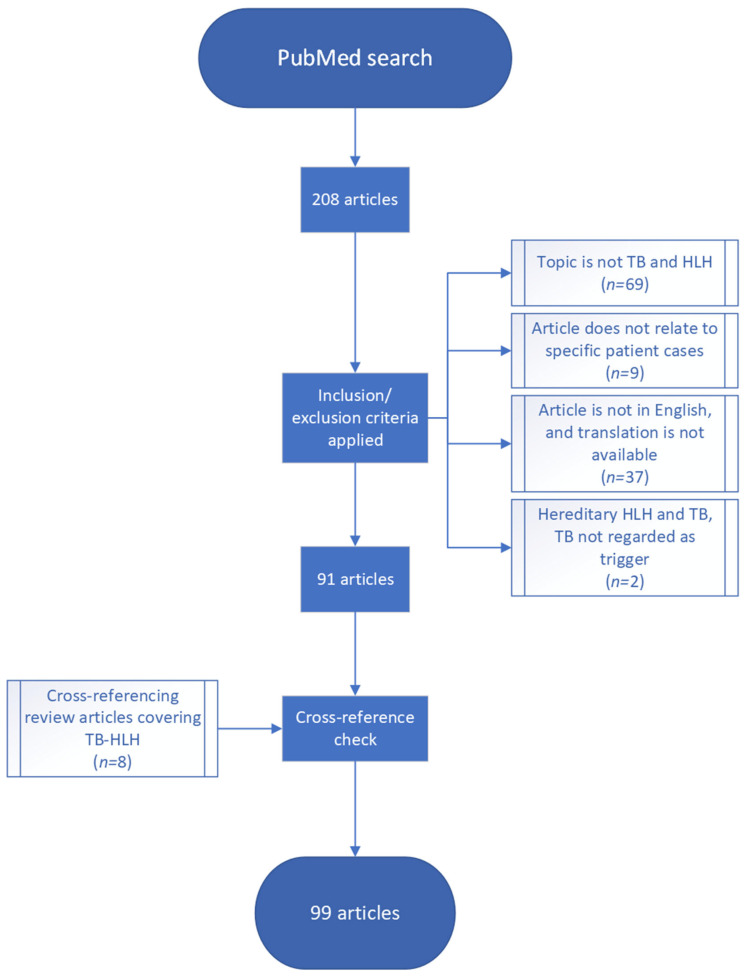
Flow chart summarizing inclusion/exclusion process (numbers in parentheses show the number of articles included or excluded due to each criterion).

**Figure 2 jcm-12-05366-f002:**
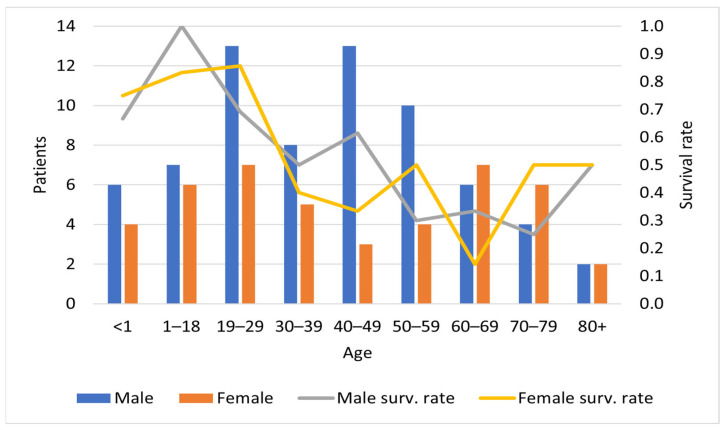
Distribution of age and gender, with respective survival rates. Patients with unknown age/gender are not included.

**Figure 3 jcm-12-05366-f003:**
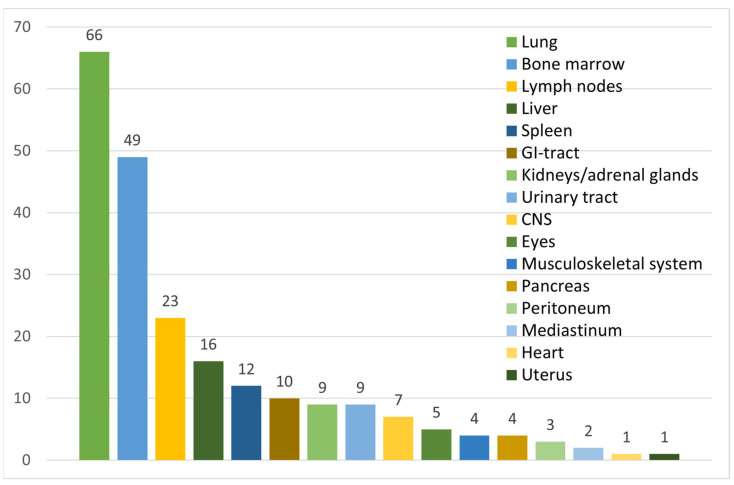
Incidence of organs in which TB was either isolated, or the organ was displaying signs of TB infection.

**Figure 4 jcm-12-05366-f004:**
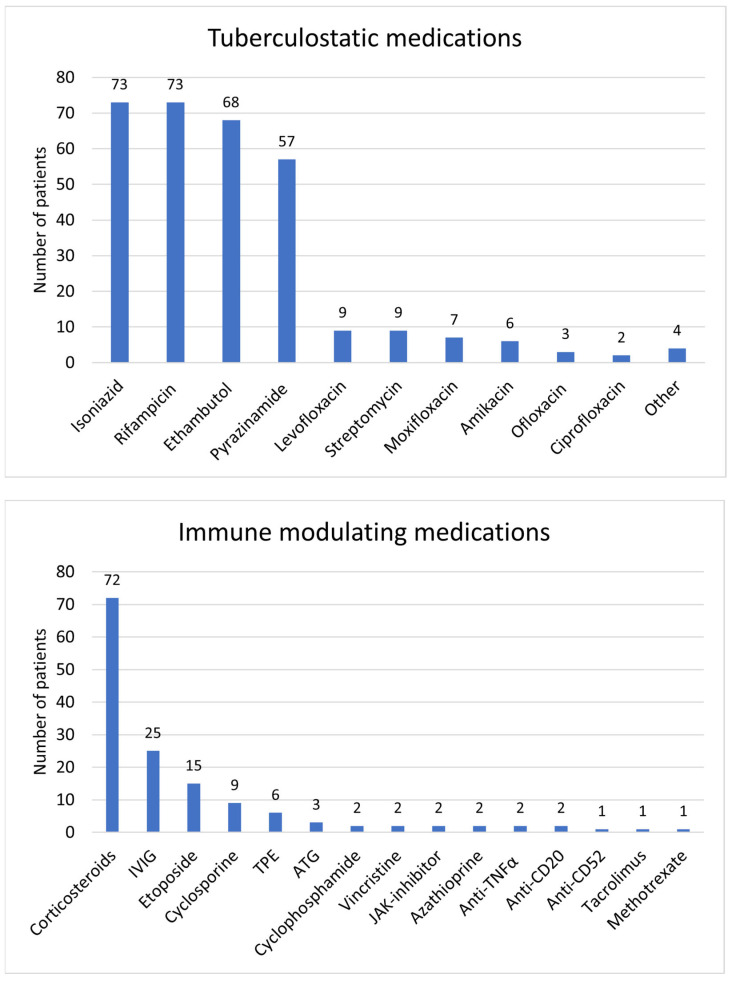
Visual distribution of medications used in the treatment of TB-associated HLH. Tuberculostatic is demonstrated in the **upper panel**, and immune-modulating agents in the **lower part**.

**Table 1 jcm-12-05366-t001:** Summary of synonyms used in the literature search.

Synonyms for tuberculosis	TB
	TBC
	mycobacterium tuberculosis
	mycobacterial infection
	miliary tb/tbc
	secondary tb/tbc
	Koch(’s) disease
Synonyms for HLH	h(a)emophagocytic lymphohistiocytosis
	h(a)emophagocytic syndrome
	h(a)emophagocytic histiocytosis
	erythrophagocytic lymphohistiocytosis
	h(a)emophagocytic lymphocytosis
	hps

**Table 2 jcm-12-05366-t002:** Clinical and biochemical findings in the study population.

	Median	Range	Number * (Percentage)
Sex			
Male			69/116 (59%)
Female			45/116 (39%)
Unspecified			2/116 (2%)
Age, years	40	0–83	
HLH-04 criteria			
Fever, °C	39	37.3–41.0	114/116 (98%)
Hepatosplenomegaly			57/96 (59%)
Isolated splenomegaly			22/96 (23%)
Isolated hepatomegaly			6/96 (6%)
Bi-/pancytopenia			103/116 (89%)
Hemoglobin, g/dL	7.8	2.4–15.5	70/90 (78%)
$\mathrm{Thrombocytes},\;\times 10^{9}/L$	37	2.5–545	77/92 (84%)
$\mathrm{Leukocytes},\;\times 10^{9}/L$	2.0	0.0–59.5	68/85 (80%)
Ferritin, μg/L	5000	370–395,644	79/83 (95%)
Triglycerides, mg/dL	292	88–777	38/58 (66%)
Fibrinogen, g/L	1.2	0.15–9.9	29/43 (67%)
Hemophagocytosis			106/116 (91%)
Soluble IL-2r, U/mL		2500–30,247	20/21 (95%)
Low NK-cell activity			13/16 (81%)
Other findings			
CRP, mg/L	107	0.9–462	36/38 (95%)
ESR, mm/hour	57	4–150	28/29 (97%)
LDH, U/L	1144	247–10,646	46/53 (87%)
Hyponatremia, mmol/L	130	123–143	17/32 (53%)
Creatinine, μmol/L	184	26.5–910	15/22 (68%)
AST, U/L	141	21–1787	51/56 (91%)
ALT, U/L	97	10–600	35/55 (64%)
Bilirubin (total), μmol/L	55	6–444	33/43 (77%)
INR	1.6	0.87–10	16/22 (73%)
Albumin, g/L	22	11–37	28/31 (90%)

* Counting how many patients fulfill HLH-04 criteria. Notably, in some of the cases reviewed, no numerical biochemical data were given—the criteria fulfilled were simply specified as such by the respective authors. For “other findings” the count represents the number of patients who exhibited pathological levels. CRP, C-reactive protein; ESR, erythrocyte sedimentation rate; LDH, lactate dehydrogenase; AST, aspartate transaminase; ALT, alanine transaminase; INR, international normalized ratio.

**Table 3 jcm-12-05366-t003:** Summary of comorbidities in the study population.

Comorbidity		Number	Percentage
No comorbidity		61	53%
Malignancy		10	9%
	Hematologic malignancy	8 *	7%
	Other malignancy	2	2%
Transplant recipient		5	4%
	Kidney	4	3%
	Liver	1	1%
Autoimmune disease		10 †	9%
Concomitant infection		9	8%
	HIV/AIDS	3	3%
	Other	6 ‡	5%
Lifestyle-associated conditions		21	18%
	Diabetes mellitus	9	8%
	Hypertension	7	6%
	Coronary artery disease	2	2%
	Aortoiliac bypass	1	1%
	Active smoker	2	2%
Pregnancy		2	2%
Renal failure		11	9%
Other	PCOS, ADS, CVA, AF, mitral insufficiency, cervical disc prolapse, hip fracture, adrenal insufficiency, alcoholism	15	13%

* 4 cases of lymphoma, 3 of leukemia, and 1 of unspecified bone marrow malignancy; † 2 cases of systemic lupus erythematosus, 2 of arthritis, 1 of granulomatosis with polyangiitis, 1 of Evan’s syndrome, 1 of sarcoidosis, 1 of Crohn’s disease, 1 of unspecified colitis, and 1 of unspecified autoimmune disease; ‡ 2 cases of Epstein–Barr virus, 1 of hepatitis C virus, 1 of malaria, 1 of schistosomiasis, and 1 of candida infection. HIV, human immunodeficiency virus; AIDS, acquired immunodeficiency syndrome; PCOS, polycystic ovary syndrome; ADS, anxiety-depressive syndrome; CVA, cerebrovascular accident; AF, atrial fibrillation.

**Table 4 jcm-12-05366-t004:** Summary of medications and supportive measures used during the course of treatment.

Treatment		Number	Percentage
Tuberculostatic			
	Rifampicin	73	63%
	Isoniazid	73	63%
	Pyrazinamide	57	49%
	Ethambutol	68	59%
	Fluoroquinolones	21	18%
	Streptomycin	9	8%
	Amikacin	6	5%
	Other *	4	3%
	Unspecified	24	20%
Cytostatic			
	Etoposide	15	13%
	Cyclophosphamide	2	2%
	Vincristine	2	2%
	Methotrexate	1	1%
Immune modulating			
	Corticosteroids	71	61%
	IVIG	25	22%
	Cyclosporine	9	8%
	Tacrolimus	1	1%
	ATG	3	3%
	JAK-inhibitor	2	2%
	TPE	6	5%
	Azathioprine	2	2%
	Anti-TNFα	2	2%
	Anti-CD20	2	2%
	Anti-CD52	1	1%
Supportive treatment			
	ECMO	2	2%
	Invasive respiratory support	28	24%
	NIV	7	6%
	Hemodynamic support	13	11%
	Renal replacement therapy	10	9%
	Transfusion	28	24%
	G-CSF	3	3%
	Antimicrobial therapy	7	6%
	Splenectomy	4	3%

* The following were only used in one patient each: rifapentine, cycloserine, trimethoprim-sulfamethoxazole, and azithromycin. IVIG, intravenous immunoglobulin; ATG, anti-thymocyte globulin; JAK-inhibitor, Janus kinase inhibitor; TPE, therapeutic plasma exchange; Anti-TNFα, anti-tumor necrosis factor alpha; ECMO, extracorporeal membrane oxygenation; NIV, non-invasive ventilation; G-CSF, granulocyte colony-stimulating factor.

## Data Availability

Data may be made available on reasonable request to the corresponding author.

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
