# Peer review of "Tuberculosis-Associated Hemophagocytic Lymphohistiocytosis: A Review of Current Literature"

_jcm, 2023, doi:10.3390/jcm12165366_

Round 1

Reviewer 1 Report

Dear authors,

Congartulations for your manuscript. It aims to perform a systematic review of hemophagocytic lymphohistiocytosis associated to TB. The article is interesting and presents a good content. However, some minor adjustments are required.

1. At line 64, in the introduction, some results are presented. I suggest that this section should end with the main and the secondary aims of the paper. Results of the current paper should be presented in the results section.

2. Also, the criteria for HLH diagnosis in table 1 and the HScore in table 2 should be presented i the methods and not in the introduction.

3. At line 198, the results are clearly described in Figure 5 but the text is quite confusing. I suggest to rewrite this paragraph detailing clearly two groups, confirmed and not confirmed diagnosis. Then, the frequency in each organ/system in each group. 

4. At line 220, an information could be misunderstood. There is a difference between tuberculosis infection (former LTBI) and active TB. IGRA is used mainly for LTBI. The exception is in children and ocular TB. So, I suggest positive IGRA results should be excluded or discussed separately. Also, the results should be presented separately

Yours Sincerely,

Author Response

Congratulations for your manuscript. It aims to perform a systematic review of hemophagocytic lymphohistiocytosis associated to TB. The article is interesting and presents a good content. However, some minor adjustments are required.

We are grateful for these positive and encouraging comments.

At line 64, in the introduction, some results are presented. I suggest that this section should end with the main and the secondary aims of the paper. Results of the current paper should be presented in the results section.

We agree with this comment, and accordegly have rephrased this section.

  1. Also, the criteria for HLH diagnosis in table 1 and the HScore in table 2 should be presented i the methods and not in the introduction.

These tables have been moved to the supplementary section, and are currently also referred to in the introduction section.

  1. At line 198, the results are clearly described in Figure 5 but the text is quite confusing. I suggest to rewrite this paragraph detailing clearly two groups, confirmed and not confirmed diagnosis. Then, the frequency in each organ/system in each group. 

We agree that this section was a little unclear written, and accordegly we have restructured this section.

  1. At line 220, an information could be misunderstood. There is a difference between tuberculosis infection (former LTBI) and active TB. IGRA is used mainly for LTBI. The exception is in children and ocular TB. So, I suggest positive IGRA results should be excluded or discussed separately. Also, the results should be presented separately. 

This section has been revised, and now includes a separate paragraph detailing the only case in which a positive IGRA was the main finding leading to the diagnosis of TB.

Reviewer 2 Report

General comments

The paper entitled “Tuberculosis-associated Hemophagocytic Lymphohistiocytosis: 2 a review of current literature” is focused on the explanation of these 2 diseses and the potential effect of TB on HL. Unfortunately, the methodology is still poorly described, such as key results, presentation is very confusing and the whole manuscript looks more like a statistical report.

Specific comments

1.     The abstract should be revised according to the following: Few study objectives, Applied method, Key values from promising results,            Also, add the novelty of this work clearly, Possible future applications of this study, Make it up to 200-250 words.

2.     Page 2, Line 61: Years included in selection should be given.

3.     Figure 2. is showing gender and age distribution of these diseases, and it must be noticed bigger incidence of male patients. Is there any specific explanation behind this phenomenon?

4.     Figures  3 and 4 should be revised, and standardized.

5.     Figures 3 and 4 should be cross-referenced, as I can not anywhere in text it is mentioned.

6.     Table 6 is too big, and confusing. Couldn’t it be summarized in some smaller table more clearly and understandable or maybe figure?!

7.     The results presentation is still very confusing and to messy. If results are presented in the table, in text should be discussed, not written it again and again.

8.     Figure 6 should be properly revised. Authors are advised to use Orgin for chart drawing. Presentation in Excel is not appropriate because it is not giving high-resolution graphs.

9.     Authors are advised to avoid reference lumping as it is present in many places.

10.  The conclusion section should be revised. Authors are advised to avoid repeating in writing, such as lines 401-406. The conclusion section should contain information as follows: Suitability of the applied method, Key values from results, Major findings and contribution, As well as limitations of the study if there is any, Possible future work, Make it up to 250-300 words.

Can be improved.

Author Response

General comments

The paper entitled “Tuberculosis-associated Hemophagocytic Lymphohistiocytosis: 2 a review of current literature” is focused on the explanation of these 2 diseses and the potential effect of TB on HL. Unfortunately, the methodology is still poorly described, such as key results, presentation is very confusing and the whole manuscript looks more like a statistical report.

We have in the revised version tried to reduce the number of tables, and moved some to suppletory to make the manuscript more readable and easier to interpret.

Specific comments

  1. The abstract should be revised according to the following: Few study objectives, Applied method, Key values from promising results, Also, add the novelty of this work clearly, Possible future applications of this study, Make it up to 200-250 words.

The journal’s author guidelines allowed a maximum of 200 words in the abstract. We have tried to restructure the abstract accordingly, highlighting the main finding in the study.

  1. Page 2, Line 61: Years included in selection should be given.

This has been revised and the timeframe included in the search is now given.

  1. Figure 2. is showing gender and age distribution of these diseases, and it must be noticed bigger incidence of male patients. Is there any specific explanation behind this phenomenon?

The paragraph discussing this point is now more clearly addressing this phenomenon (line 229-239).

  1. Figures  3 and 4 should be revised, and standardized.

We agree that these figures overlapped and actually are present in Table 2, accordingly we have removed these figures in our revised version of the manuscript.

  1. Figures 3 and 4 should be cross-referenced, as I can not anywhere in text it is mentioned.

As stated in the previous paragraph these figures are removed from the revised version of the manuscript.

  1. Table 6 is too big, and confusing. Couldn’t it be summarized in some smaller table more clearly and understandable or maybe figure?!

We agree that this Table is a lite larger and could be hard to interpret. Although, we believe this information is highly important for the paper as a whole, and for the background for HLH induced TBC. Accordingly, we have removed this table to supplementary section of the article.

  1. The results presentation is still very confusing and to messy. If results are presented in the table, in text should be discussed, not written it again and again.

We have tried to restructure this and avoid unnecessary repetitions. Specific discussions of findings are placed in the discussion part of the manuscript.

  1. Figure 6 should be properly revised. Authors are advised to use Orgin for chart drawing. Presentation in Excel is not appropriate because it is not giving high-resolution graphs.

We have revised this figure, and it should now be more easily readable.

  1. Authors are advised to avoid reference lumping as it is present in many places.

We have tried to restructure this in the revised version. References have been more spread out in the places where this is appropriate.

  1. The conclusion section should be revised. Authors are advised to avoid repeating in writing, such as lines 401-406. The conclusion section should contain information as follows: Suitability of the applied method, Key values from results, Major findings and contribution, As well as limitations of the study if there is any, Possible future work, Make it up to 250-300 words.

We have revised and restructured the conclusions section accordingly.

Reviewer 3 Report

The article design is well done. It was planned to determine the relationship between tuberculosis disease and Hemophagocytic Lymphohistiocytosis syndrome. The literature review was done in detail. Due care was taken in the selection and exclusion of literature. Tables that support the literature have been created in detail. It is planned to increase the knowledge and awareness of Hemophagocytic Lymphohistiocytosis syndrome among readers. However, due to the small number of patients, interest among readers is likely to be somewhat low. 

Author Response

We are grateful for these mainly positive comments. We agree that the number of patients is low, although given the rarity of this disease, we believe a comprehensive review of this kind is important for physicians meting patients with HLH and TBC. Hence, we believe the paper is of interest for the upcoming special issue in Journal of Clinical Medicine.

Round 2

Reviewer 2 Report

After review article quality is improved and could be considered for publication.

Minor editing changes are required.